# Acoustic Wave Reflection in Water Affects Underwater Wireless Sensor Networks

**DOI:** 10.3390/s23115108

**Published:** 2023-05-26

**Authors:** Kaveripakam Sathish, Monia Hamdi, Ravikumar Chinthaginjala Venkata, Mohammad Alibakhshikenari, Manel Ayadi, Giovanni Pau, Mohamed Abbas, Neeraj Kumar Shukla

**Affiliations:** 1School of Electronics Engineering, Vellore Institute of Technology, Vellore 632014, India; 2Department of Information Technology, College of Computer and Information Sciences, Princess Nourah bint Abdulrahman University, P.O. Box 84428, Riyadh 11671, Saudi Arabia; 3Department of Signal Theory and Communications, Universidad Carlos III de Madrid, 28911 Leganés, Madrid, Spain; 4Department of Information Systems, College of Computer and Information Sciences, Princess Nourah bint Abdulrahman University, P.O. Box 84428, Riyadh 11671, Saudi Arabia; 5Faculty of Engineering and Architecture, Kore University of Enna, 94100 Enna, Italy; 6Electrical Engineering Department, College of Engineering, King Khalid University, Abha 61421, Saudi Arabia

**Keywords:** reflection, acoustic waves, underwater environments, incidence, permeability

## Abstract

The phenomenon of acoustic wave reflection off fluid–solid surfaces is the focus of this research. This research aims to measure the effect of material physical qualities on oblique incidence acoustic attenuation across a large frequency range. To construct the extensive comparison shown in the supporting documentation, reflection coefficient curves were generated by carefully adjusting the porousness and permeability of the poroelastic solid. The next stage in determining its acoustic response is to determine the pseudo-Brewster angle shift and the reflection coefficient minimum dip for the previously indicated attenuation permutations. This circumstance is made possible by modeling and studying the reflection and absorption of acoustic plane waves encountering half-space and two-layer surfaces. For this purpose, both viscous and thermal losses are taken into account. According to the research findings, the propagation medium has a significant impact on the form of the curve that represents the reflection coefficient, whereas the effects of permeability, porosity, and driving frequency are relatively less significant to the pseudo-Brewster angle and curve minima, respectively. This research additionally found that as permeability and porosity increase, the pseudo-Brewster angle shifts to the left (proportionally to porosity increase) until it reaches a limiting value of 73.4 degrees, and that the reflection coefficient curves for each level of porosity exhibit a greater angular dependence, with an overall decrease in magnitude at all incident angles. These findings are given within the framework of the investigation (in proportion to the increase in porosity). The study concluded that when permeability declined, the angular dependence of frequency-dependent attenuation reduced, resulting in iso-porous curves. The study also discovered that the matrix porosity largely affected the angular dependency of the viscous losses in the range of 1.4 × 10^−14^ m^2^ permeability.

## 1. Introduction

### 1.1. Basic Discussion of UWSN

Wireless Sensor Networks (WSN) have vast application potential in environmental and military monitoring. Wireless sensor nodes, which are often located at a considerable distance from the Base Station (BS) and other nodes, are responsible for data collection, storage, and analysis. The nodes can be built in one of two ways: with the sensor node placements and communication topologies already planned out; set up without any planning [1]. Because of their capacity for random deployment, sensor nodes are crucial for providing relief during natural disasters and for other applications that require a rapid setup. There is an impediment to communication between the sensor nodes and the high-power BS, also known as the sink. They have no face-to-face interactions with their local peer group; instead, everything is carried out via broadcasts. The sensor network can dynamically adapt to alterations in the network’s topology, whether those alterations originate from the removal of nodes or from the installation of new nodes. The expansion of the sensor network to include several new nodes is warranted as a result. If one of the nodes fails, the others will automatically reorganize themselves [2].

When a sensor node in a WSN collects fresh data, it promptly sends them to the node directly to its left. Each node discovers and exchanges information with its nearest neighbor to achieve this. The resulting reduction in power usage during data transmission is a desirable byproduct. Following the discovery phase, the data transfer phase comprises the relaying of information between the closest sensor nodes (also known as the phase of locating the nearest node). A single hop is used to accomplish this, which implies that one sensor node immediately transfers the information it has gathered to another sensor node. The third and last step is to send these data to a BS or sink node. Since the year 2000, significant progress has been made in the fields of underwater network protocols, modems, and communication for Underwater Wireless Sensor Networks (UWSNs) [3,4]. Small, low-powered, and power-hungry sensors are set at various depths in the water to track occurrences beneath the surface. These nodes can take data from sensors, process them, and then communicate them to a surface sink via acoustic waves.

A network of sensors is typically installed aboard Autonomous Underwater Vehicles (AUVs) or Unmanned Undersea Vehicles (UUVs) to monitor the ocean floor. Due to radio waves’ poor performance, acoustic signals are the favored alternative for underwater communication. Acoustic communication problems, in comparison to electromagnetic waves, include, but are not limited to, greater error rates, lengthened propagation delays, lower bandwidths, and transmission disruptions [5]. Sensor nodes for underwater communication are made up of a range of different components. These components comprise the sensor unit, the communication unit, the power unit, the acoustic modem, and the processing unit. The sensor gadget receives environmental factors such as temperature and pressure from its surroundings. The Radio Frequency (RF) signal is converted into audible tones by the acoustic modem [6].

It is general knowledge that the deployment of nodes in UWSNs is the essential activity that contributes to the control of topology, as well as the provision of network services, boundary conditions, and routing. When classifying the deployment methods, they are placed into one of these three categories: (i) static deployment, (ii) self-adjusted deployment, and (iii) movement-assisted deployment. The goal of the deployment, the types of nodes, the amount of energy that is utilised, and the level of computational complexity are all taken into consideration. Once the nodes have been positioned in a static deployment, it is not possible to relocate them in any way [7]. The essential assumptions behind this technique are the building blocks for a range of various deployment tactics. The data that are gathered by the sensor nodes in this setup are sent upwards, while the nodes themselves are placed in a bottom-based cluster design. This is achieved through a chain of hops in the intermediate range [8]. Each node in a deployment that employs self-adjustment is capable of modifying its own position within the network. By lowering the incidence of the overlap scenario and enhancing the link quality, this deployment technique increases network connectivity.

It is generally believed that routing is the fundamental challenge in any network. Routing protocols are responsible for identifying and maintaining routes. Although research on underwater acoustic communication has been conducted for more than a decade, this field of study is still in its infancy. As a result, the most significant protocols for UWSN to date are analysed, together with their respective advantages and disadvantages. A recent study has revealed that the majority of routing algorithms employ geographical considerations. A hybrid system is made up of both stationary and mobile sensors that function together. In 2D networks, static sensor nodes often reside on the seafloor and communicate with sink nodes via multi-hop communication across separate clusters [9]. This enables the nodes to communicate with sink nodes. Sensors in a static 3D architecture can also be placed at different depths with the help of inflatable buoys by altering the length of the wire coming up from the sea floor. This helps to obtain the intended outcome. Sensor nodes in a mobile architecture are not tied to one another and are free to wander, allowing the network topology to change in real-time. The use of two transceivers is required to realize the mobile node’s full data-gathering network capabilities. This category could include sea gliders, autonomous underwater vehicles, and remotely operated underwater vehicles. Third, a hybrid system employs both stationary and mobile sensor nodes to perform specific tasks [10]. This is due to the fact that a hybrid design incorporates the best of both worlds. In hybrid networks, mobile nodes act as controllers or routers, coordinating data collection from traditional sensors that are fixed in place. With the use of UWSNs, ocean monitoring and forecasting may be able to achieve a better level of precision. As a result, it is necessary to coordinate the software with the hardware in order to achieve the latter purpose.

### 1.2. Preliminary Description of Acoustic Wave Reflection in UWSN

Researchers discovered that autonomous underwater vehicles and underwater sensors can collaborate to perform observation and monitoring activities at varying depths. Adaptive sampling and self-configuration are said to be impossible to achieve without a network coordination approach that allows for the sensor and AUV to be combined, as stated in [11]. By moving and repositioning mobile vehicles along prescribed patterns, adaptive sampling provides a more thorough coverage of a region during data-gathering. Self-configuration is a process used by autonomous underwater vehicles to identify network gaps when sensor nodes fail or channels are lost. It is critical to design pathways that eliminate barriers so that sensor nodes and autonomous underwater vehicles can collaborate efficiently.

Once a wave comes across a variation in medium sideways, its route of propagation, partial reflection, and transmission occur, as depicted in Figure 1. The reflection coefficient is defined as the ratio between the amplitude of the reflected wave and the incident wave amplitude. When traveling, acoustic waves encounter an interface between two materials with substantially different acoustic properties; the waves are both reflected and perturbed by the solid surface [12]. As shown in Figure 2, a perpendicular wave is produced in the near field when a reflected traveling wave interacts with an incident wave. This occurrence can also be observed in the far distance. This situation results in a doubling of the local pressure due to the near-field particle velocity becoming zero [13].

Due to a phenomenon known as attenuation, the reflected wave’s intensity will decrease as its detachment, starting the reflective medium, rises [14,15,16,17]. When the control of the wave that is reflecting is less than the power of the wave that is incident, interference is reduced. As interference reduces, the phase mismatch between the sound pressure and particle velocity decreases as a wave travels further into its far field. This situation occurs due to the constancy of sound pressure [18].

In recent decades, scientists have paid considerable attention to the phenomena that occur when an acoustic plane wave rebounds off a fluid–solid contact. These hypotheses can be traced back to Biot’s seminal papers [19,20]. Several models have been created to study the lossless scenario; however, these models fail to account for the impacts of dissipation and thermoviscous resistance. In recent years, scientists have incorporated thermo-viscous losses into models to explore the decrease in the reflection coefficient.

Wave reflection at fluid–solid interfaces is a significant research area for Non-Destructive Testing (NDT) and material characterization [21]. According to the author, the most significant contribution of this study to the field is the assessment of the acoustic attenuation response to variations in solid permeability and porosity. This is something that has never been thoroughly considered in the past. This work intends to fill in some of the gaps found in prior research by undertaking an intensive comparison analysis of the numerous models that employ different propagating media and, as a result, the comparative significance of these features for a wide range of parametric combinations [22,23,24,25].

Important measuring parameters that show the acoustic performance of underwater materials such as anti-acoustic baffles and anechoic tiles include the reflection and transmission coefficient of the sample under the plane wave that is perpendicular to the incidence. As a consequence of this, it is essential in the field of underwater acoustic engineering to accurately calculate the complicated reflection and transmission coefficient of underwater materials. The useable frequency range of underwater acoustic materials is expanding to encompass lower and lower tones as maritime technology continues to make strides forward. When a material is first being designed, the acoustic performance of the material is evaluated primarily by the use of the test results obtained from testing tiny samples inside an acoustic tube [26]. The acoustic tube measuring approach may be broken down into its component parts, the first of which is the control of the sound source at one end of the acoustic tube. Because of this, it is possible for the acoustic tube to generate plane waves at frequencies lower than the first-order cut-off frequency. From the perspective of the sound field that is produced inside the tube, the acoustic tube method, which is also known as the impedance tube method, can be subdivided into three different sub methods: the standing wave tube method, the pulse tube method, and the Travelling Wave Tube (TWT) method. 

The pulse tube method is one such technology that excels at high frequencies. When conducting frequency-dependent testing, as the frequency of the measurements lowers, a longer acoustic tube is required. As a result, measuring the reflection and transmission coefficients of materials at low frequencies can be problematic. Only the acoustic reflection coefficient may be detected when studying the acoustic properties of materials in a standing wave tube, and the acoustic impedance value of the sound tube’s downstream must be verified [27]. The sample is placed in the centre of the acoustic tube, the primary sound source is upstream of the tube, and the secondary sound source is downstream of the tube to absorb transmission-reflection sound waves, in accordance with the TWT method for determining the reflection and transmission coefficients of a material. The sound wave that flows through the acoustic tube is created by a main source. Plane-moving waves are produced by primary sound generators. In an ideal world, the acoustic tube’s length would be infinite because the travelling wave field can be formed in the tube’s under-material half. The TWT method can be used to obtain both the low-frequency acoustic reflection and transmission coefficients of materials at the same time.

Either passive or active control can be applied in the management of TWT. Passive control requires the addition of a substance that absorbs sound after the acoustic tube. This ensures that the system absorbs the sound waves created by the transmission-reflection process. However, because the efficacy of the sound-absorbing material decreases with decreasing frequency, there will be reflected sound in the right portion of the tube. If active control is employed, the relationship between the secondary sound source and its surface reflection coefficient [28] must be determined. This will occur if the active control mechanism is implemented. After this stage has been completed, adjustments can be made to the loading signal for the secondary sound source to account for variations in the reflection coefficient. It is feasible to construct a travelling wave field once the reflection coefficient reaches a crucial value. Even while this technology has the ability to generate a travelling wave field in a sound tube with pinpoint accuracy, it has a number of limitations, the most notable of which are the complex computation and the lengthy control procedure. A little time will be required for the creation of a travelling wave field. As a direct result, very few companies currently utilise this strategy.

By analyzing the fundamental characteristics of the curves used to characterize the reflection coefficients, this study aims to gain insight into how various material components and propagation media affect the acoustic signature [29]. Among others, acoustic levitation, non-destructive testing, medical imaging, and sonar could benefit from the findings of this investigation. The protection of electronic equipment from electromagnetic interference and human tissues from excessive radiation is an active area of study and development in this sector [30]. This investigation focuses on a frequency range in which the wavelength of the disturbances grows closer and closer to that of the micro-defining structure’s dimensions.

Underwater wireless sensor networks can be used to monitor the environment and explore the core of the earth. The phenomena of acoustic wave reflection in water may have a significant impact on each of these domains. In the context of environmental monitoring, UWSNs can be used to measure water quality, temperature, pressure, and other environmental parameters. However, because of the existence of reflecting surfaces in the water, signal distortion, attenuation, and multipath interference may occur. These occurrences may impact the data’s correctness and dependability [31].

Acoustic wave reflection can be researched, and signal processing algorithms can be created to counteract its negative impacts on UWSNs. Researchers can, for example, employ simulations or experimental investigations to establish the ideal placement of sensors and the most effective signal processing algorithms to limit the impacts of reflection and increase data accuracy. The quality of the collected data is critical to reaching this goal. In the field of geological exploration, UWSNs are a useful tool for investigating rock formations, sediments, and other geological features. The existence of reflecting items in the water, however, limits the depth to which the water may be analysed, which can impair the trustworthiness of the gathered data [32].

Researchers can overcome these obstacles by studying the impact of acoustic wave reflection on underwater sensor networks. Using these tools and strategies can improve the precision and extent of the exploration. Scientists, for example, can design sensors that detect and analyse the intensity of reflected acoustic waves to provide more exact data on the geological properties under study. The global effects of acoustic wave reflection in water can have a considerable impact on the use of underwater wireless sensor networks for environmental monitoring and geological research. Researchers can improve the accuracy and reliability of the obtained data and expand the usefulness of UWSNs in various industries by examining these phenomena and developing new technologies and strategies to mitigate their impact.

### 1.3. Problem Formulation

Prior to this research, the author was unaware of any extensive study on how differences in solid permeability and porosity affect sound attenuation. This is the paper’s primary contribution to the study of this topic. This study aims to close these gaps by conducting a complete comparative analysis of numerous models employing different propagation media and, as a result, the relative relevance of these attributes for various parametric combinations. The existing literature considers limiting scenarios for the aforementioned media to address the impact of particular material properties. This study, which focuses on the impact of specific material properties, considers limiting scenarios for the aforementioned media. Analyzing the key aspects of the reflection coefficient curves for different propagation mediums and material factors can help to understand the impact material properties have on the acoustic signature. The findings of the study could shed significant new light on the process of wave propagation. Acoustic levitation, non-destructive testing, medical imaging, and sonar are just a few of the potential applications for this information. This area of research is currently investigating the difficulties associated with insulating electronic devices from electromagnetic interference and biological tissues from high radiation. The research focuses on a frequency range where the wavelengths of the disturbances are comparable to the average lengths of the microstructure.

### 1.4. Proposed Solution

To the best of the author’s knowledge, this is the first time anyone has tested the acoustic attenuation response to changes in solid permeability and porosity. This work seeks to fill some of the gaps found in prior research by conducting a thorough comparison analysis of the many models that use different propagating media and, as a result, the comparative significance of these properties across a wide range of parametric combinations. The goal of this study is to gain knowledge about how various material properties and propagation media influence the acoustic signature by undertaking an examination of the fundamental aspects of the reflection coefficient curves. The findings of this investigation may not only shed light on wave propagation characteristics, but may also be valuable in a wide range of applications, including acoustic levitation, non-destructive testing, medical imaging, and sonar. The protection of electronic equipment from electromagnetic interference and of human tissues from excessive radiation is a topic that is being actively explored and developed further within this field. The frequency range under consideration is one in which the wavelength of the disturbances approaches that of the size of the micro-defining structure.

The planned study, titled “Acoustic Wave Reflection in Water Affects Underwater Wireless Sensor Networks”, will have a significant impact on how we think about underwater communication and networking in the future. UWSNs have emerged as a promising solution in light of the increased demand for underwater remote monitoring and data collection. Acoustic waves are the primary communication channel for UWSNs; nevertheless, the propagation of these waves underwater creates a number of obstacles. The proposed study aims to investigate and solve the problem of acoustic wave reflection, which contributes considerably to signal degradation and loss in UWSNs. The proposed research aims to gain a better understanding of how acoustic wave propagation is affected by different materials such as sediments, rocks, and biological structures, and how this, in turn, affects the performance of underwater wireless sensor networks by investigating the behaviour of acoustic waves at the interface between water and different materials such as sediments, rocks, and biological structures. The proposed research not only advances the existing state of the art in underwater acoustics and UWSNs, but also provides a fresh method for resolving the issues presented by UWSNs. The use of poroelastic materials in the design of acoustic barriers to reduce acoustic wave reflection is a novel notion that has received little attention in this area. This technology has the potential to considerably improve the performance of UWSNs by minimizing the amount of data loss and improving the extent to which communication can be relied on. The proposed research is particularly relevant to the current state of the art in underwater communication and networking because it takes a novel approach to tackle the issues provided by UWSNs. The proposed research is really important. The findings of this study could have a substantial impact on the future of UWSN design and deployment, potentially leading to more efficient and trustworthy underwater communication systems.

The main contribution of the manuscript is highlighted as follows:Development of a design model for UWSN’s water–sediment interaction, taking into consideration the effects of acoustic reflections;A comparison of the simulation’s results with the model currently in use;An investigation on the permeability and porosity of the design model.

The remainder of this paper is organized as follows. In Section 2, we discuss the modelling procedure utilized for the geometries. The conditions at the boundaries are presented in Section 3. Section 4 is devoted to the presentation and discussion of the findings with the application of various input elements to each case. In Section 5, the closing considerations are discussed.

## 2. Modeling Geometry

This section will provide the considered model geometry for the design with different parameters. We used Stoll and Kan’s mathematical model to describe the geometries shown in Figure 3 and Figure 4 with the proper boundary conditions.

This confirmation model computes the acoustic wave reflection coefficient in the context of a water–sand interaction [33,34]. Wave reflection and transmission occur at the water–sediment interface for homogeneous waves originating in the fluid domain. These waves travel in a straight line from the fluid domain to the water–sediment interface [35]. Biot’s theory is applied to the generation of a model of the sediment domain via the Poroelastic Waves interface. This technique considers the complex interplay between pressure waves in the watery fluid and elastic waves in the porous matrix [36,37].

Figure 4 depicts the use of COMSOL for Geometrical Modelling. A strong understanding of plane wave reflection and refraction coefficients is essential to interpreting the acoustic data utilized in marine seismology appropriately [38]. These kinds of models are helpful for understanding the processes beneath the surface and the simulation design model, as shown in Figure 5. 

In two dimensions, Floquet periodic functions are utilized to model the problem. Conditions in the porous and acoustic domains are based on the condition of both domains; to ensure that the problem is solved, we may make the domain size quantitatively dependent on light wavelength in water (therefore, there will be several occurrences per unit of time). The wavelength (represented by the letter) is also the width of the computational unit cell (represented by the letter W).

The height H is obtained by multiplying the wavelength by two. Regarding the domain in which computing occurs, the fluid and porous domains are shortened by utilizing layers compatible with one another: Perfectly Matched Layers (PMLs). The optimal thickness is selected to be one-third of the wavelength. 

Table 1 summarizes the several materials employed in the solid domain as well as the numerous input parameters that were examined. The parts that follow the specific cases are discussed in greater detail. The modelling of frequency-dependent geometry is made possible by metric sweeping through numerous input parameter values that are considered for the driving frequency. To create a very small mesh at the interface, the porous domain mesh was constructed with a single boundary layer mesh element. The same mesh was used to model both the source and the destination in order to mimic the periodic conditions using the Floquet–Bloch periodicity.

Biot’s theory is expanded to account for the viscoelastic features of the porous matrix skeleton to simulate sediment. The bulk modulus and the shear modulus can contain a small imaginary component, allowing this to be accomplished (a constant attenuation factor). Three types of waves can pass through the porous material: shear waves, quick pressure waves, and slow pressure waves. As a result of the interplay between these waves and the incoming homogenous pressure wave, the fluid reflection coefficient behaves in a manner that is not trivial [39].

Sound waves are an excellent example of how wave energy may be altered as it travels through a solid. Obliquely incident longitudinal waves allow for particles to move in a transverse direction, generating shear (transverse) and Rayleigh waves at the interface. When the acoustic impedances of two materials differ, mode conversion occurs, causing a disturbance in the pressure and potential field [40]. The wave will be refracted at the border between the two substances in proportion to the difference in acoustic velocity. Because the longitudinal wave moves faster than the shear wave, it has a greater refractive index. When an electromagnetic wave collides with a horizontally flat surface, it splits into reflected and transmitted components [41].

The incident pressure of an acoustic wave in a UWSN refers to the pressure exerted by the wave on the sensors. The pressure exerted by an acoustic wave in water is dependent on the frequency of the wave and the distance from the source. At low frequencies, the pressure exerted by the wave decreases with increasing distance, while at high frequencies, the pressure can increase with distance due to focusing effects, as shown in Figure 6. To calculate the incident pressure of an acoustic wave in a UWSN, we can use the following formula, represented in Equation (1):(1)P=ρcΔf
where P is the pressure of the acoustic wave, ρ  is the density of the water, c is the speed of sound in water, and Δf is the frequency bandwidth of the wave.

The reflected pressure of an acoustic wave in a UWSN refers to the pressure of the wave that bounces back from an object or boundary in the water, such as the seafloor, a rock, or a submerged structure, and returns to the sensors in the network. The reflected pressure can affect the quality of the acoustic signal in the UWSN and can create interference and noise in the communication channels, as shown in Figure 7.

The total pressure of an acoustic wave in an UWSN refers to the sum of the incident pressure and the reflected pressure of the wave at any given point in the network. This total pressure is important because it affects the energy consumption and performance of the sensors in the UWSN. The energy required to detect and decode the communication signals is directly proportional to the total pressure of the acoustic wave, as shown in Figure 8.

The displacement of an acoustic wave in an UWSN refers to the distance that the water particles move back and forth as the wave propagates through the water. As the acoustic wave travels through the water, it creates a series of compressions and rarefactions that cause the water particles to vibrate back and forth around their equilibrium positions, as shown in Figure 9.

The speed of acoustic waves in an UWSN is an important parameter that affects the propagation and communication of the waves in the water. The speed of sound in water is affected by various environmental factors, such as temperature, salinity, and pressure. The typical speed of sound in seawater is around 1500 m per second (m/s), which is about four times faster than the speed of sound in air. The acoustic wave speed in UWSN is 1584.73 (m/s) in the simulation model, which is nearly the theoretical, as shown in Figure 10.

The fluid domain is represented by classical pressure acoustics, which requires lossless solutions to the equations. Helmholtz came up with this equation. Biot’s theory, which is a sediment domain model, can answer both difficulties. The pressure and displacement fields act simultaneously on the porous matrix. The key is Multiphysics integration. The Multiphysics Acoustic–Poroelastic Waves Interaction interface is used to build the problem [42].

## 3. Conditions at the Boundaries

This study looks at the oblique incidence of homogenous waves from a fluid medium on different surfaces [43]. Water interactions with sediment, stainless steel interactions, ceramic interactions, air interactions with sediment, and air interactions with solids were all modeled using COMSOL Multiphysics 5.6.

Table 1 summarizes the several solid domain materials and input factors considered. The sections that relate to the various illustrations offer additional clarification. By executing a parametric sweep across various input parameters that influence the driving frequency, it is possible to simulate frequency-dependent geometries [44]. The porous domain mesh was built with a single mesh element for the boundary layer, resulting in an excellent mesh at the interface. Since Floquet–Bloch periodicity was used to characterize the periodic conditions, the same mesh was applied to model both the source and the destination [45,46].

COMSOL Multiphysics 5.6 was used to execute a frequency domain simulation with the pressure acoustics module [47]. This occurrence aimed to put the model to the test. Using the Pressure Acoustics node, we incorporated the frequency-domain equations into our time-harmonic and Eigen frequency acoustics model. In a half-space at z = 0, a material with variable porosity and permeability was represented [48]. 

Green [49] stated in this hypothesis, supported by experimental evidence, that the intensity of the reflection coefficient never reaches zero but rather reaches a minimal value. He accomplished this by claiming that the reflection coefficient is always positive. He concluded that the pseudo-Brewster angle anomaly at the Rayleigh angle was caused by dissipation, resulting from ionic conductivity in highly refractive materials. He also noticed that the minima rose fast with the index of refraction, resulting in a significant reflection coefficient at this angle for highly refractive material.

Becker and Richardson [50] established that shear wave attenuation per wavelength directly affects the depth of the reflected amplitude minimum. This was the conclusion they reached. Using parallel and perpendicular polarisation, they discovered that shear wave attenuation was the dominant factor dictating the depth of the Rayleigh critical angle reflection minimum, whereas velocities (primarily shear) determined the angular position of the minimum. This was demonstrated by their studies, which revealed that the Rayleigh critical angle reflection minimum was shallower than expected [51]. If a plane wave radiates homogeneously through the air and is incident on an air–solid interface, the projection of the actual propagation vector onto the interface will be equal to the projections of any waves formed at the interface, according to Snell’s law. This occurs when a plane wave strikes an air–solid interface [52].

A free tetrahedral-based preconfigured mesh with the normal setting is utilised so that both the design assembly and the surrounding air can be depicted accurately. We make use of a sweeping mesh that has six different layers in the PML layer [53]. As a direct consequence of this, there are a total of 24,304 elements, which have a combined total of 123,502 degrees of freedom. According to the results of this meshing, the regions of air and dielectric material that are located next to the copper radiating components will have the largest concentration of elements of the respective convergence rate [54,55]. Figure 11 represents the mesh view of the current design model. 

## 4. Results and Discussion

The results of computational investigations on the impact of material qualities on the reflection of flat fluid waves are described here. Their impact on the critical angle phenomena and the minimum reflection coefficient are particularly examined. The constitutive parameters for this study were chosen using suggestions from the available literature on the water–sediment interaction [56]. Choosing the permeability, porosity, and driving frequency while fully appreciating the acoustic response via reflection coefficient curve analysis is a complex undertaking. Furthermore, because viscous and thermal losses were taken into account in the solid domain modeling used in this study, the materials chosen for the following scenarios sought high damping and attenuating qualities while also representing a diverse collection of porosities and permeabilities [57].

The ratio of reflected to incident acoustic energy at the water–sediment interface is the reflection coefficient of an underwater wireless sensor network (UWSN). It regulates the amount of energy available for acoustic signal transmission and reception, which has an impact on UWSN performance. The reflection coefficient is determined by the properties of the water and sediment, the frequency of the acoustic waves, and the angle of incidence. Acoustic waves with steeper incidence angles and higher frequencies have higher reflection coefficients. The impedance mismatch is determined by the acoustic properties of the two layers. UWSN reflection coefficients can be measured using both experimental and numerical methods. For experimental measurements, echo sounders, acoustic transducers, and sonar systems are utilised. The reflection coefficient as a function of frequency and angle of incidence can be simulated using finite element or finite difference methods. The reflection coefficient must be considered in the design and optimization of UWSN systems. To optimize acoustic transducers and other components for UWSN performance, the reflection coefficient, permeability, angle of incidence and absorption coefficient must be carefully estimated.

Metals, ceramics, and composites are some examples of modern materials that are utilised in the manufacturing industry and other related fields. As a direct consequence of this, the following chemicals will be researched throughout the course of this study: Composites and other unaccounted-for materials are modelled as poroelastic solids with varying porosities and permeabilities. Some examples of these types of materials include silicon carbide, stainless steel, sand sediment, and others. Experiments were conducted at these starting points in order to gain a better understanding of the role that each parameter plays in determining the dynamic response of the system. The modelled system has a periodic characteristic, which is depicted in Figure 12. The reflection coefficient as a function of the angle of incidence is depicted in Figure 13.

A simulated experiment showed that the water–sediment interface is responsible for the reflection of plane-wave radiation. In order to imitate the properties of sand, the solid was modelled as a poroelastic sediment, and a low-frequency analysis was carried out in order to match the values that Stoll and Kan decided upon. The findings of Stoll and Kan’s research demonstrated that the reflection coefficient shifts in value depending on the angle of incidence and the frequency being considered. This model is shown to be realistic when there is a high level of concordance between the outcomes of the simulation and those that have been published. Figure 14 is a representation of the reflection coefficient as a function of the driving frequency. The relationship between the absorption coefficient and the angle of incidence is shown in Figure 15.

For instance, at low frequencies, the absorption coefficient is typically dominated by viscous losses, which are related to the viscosity of the medium. The abnormal behavior could be due to the specific properties of the material being used in the experiment or simulation, which may have a nonlinear response at low frequencies. It is also possible that the abnormal behavior is related to the angle of incidence of the wave. At certain angles, the wave may interact with the material in a way that enhances or reduces the absorption coefficient, as observed in Figure 15.

Figure 16 depicts the overall pressure, while Figure 17 displays the displacement at various frequencies. Both figures are given below. According to the findings of this study, the reflection of plane waves at the interface between a fluid and a solid is particularly sensitive to both the propagation medium and the poroelastic solid model. The investigation into the nature of the two people’s connection led to the discovery of these findings. Both attenuation and losses have an effect on the patterns of reflection, which, in turn, have an effect on the curve that is ultimately created. Plots of the reflection coefficient curves were created by striking a plane wave radiation at an angle on a number of different interfaces.

Researching how the minima of the reflection coefficient and the Rayleigh/Shear crit-ical angle for the air–solid interface vary depending on the features of the solid material may provide additional insight into the particular influence that these components have on the acoustic response. At the point where the air and the solid meet, the pseudo-Brewster angle phenomena was found to exist for a diverse set of material properties. This was identified at the interface between the air and the solid.

The reflection of acoustic waves in water affects wireless sensor networks that are deployed underwater, and comparisons with other models already in use are possible. It is beneficial to compare the results of many models in order to determine the most accurate and trustworthy technique for modelling the behaviour of UWSNs in various scenarios. This can be performed by comparing the results of different models. A significant number of models have been built to investigate the effect of reflection in water on acoustic waves when employed in UWSNs. These models differ in the assumptions they make and the mathematical formulations they employ to express those assumptions, which can be based on actual data or numerical simulations. In certain models, ray-tracing techniques are used to simulate acoustic wave propagation across layers of water and sediment; however, in others, analytical solutions based on the two-layer fluid model or the waveguide theory are used instead. These models’ accuracy and performance can be compared across a wide range of variables, including the amount of water present, the type of silt, and the acoustic frequency. The angle of incidence, frequency, and polarisation of the waves, as well as the properties of the water and sediment layers, all influence acoustic wave reflection in water. These elements can be weighed against one another to evaluate their relative importance. By comparing the results of multiple models, it is feasible to construct models that are more accurate and robust by modelling the behaviour of UWSNs. This will allow one to gain a better knowledge of how acoustic wave reflection affects the performance of UWSNs.

## 5. Conclusions

According to the findings of this study, the propagation medium and the poroelastic solid model are sensitive. These conclusions were discovered by examining the refraction tendencies change depending on the characteristics of the propagating wave and the curve created as a result of losses and attenuation. The mathematically sound character of the model requires that the shape of the periodic unit cell be made to rely on the wavelength for all investigated frequencies. This occurrence suggests that the geometry is subject to frequency-dependent change. A parametric sweep that incorporates a geometry change in the entire process is necessary to simulate this with f0 as the starting frequency, also known as the “sweep parameter”, and can be used as the research frequency in the Frequency Domain analysis. At first, the PMLs do not automatically correct for the slanted angle at which the incandescent lights generate the waves. Planar waves impacting at an angle on the surface will yield the appropriate effect when a lower PML stretching factor is required, such as when 1/cos (theta) is utilized. It has been suggested that a wave with an incidence angle that is more significant than the critical angle of total internal reflection be used in sedimentary areas with a polynomial extension. This occurrence indicates that the organization in the porous matrix is haphazard. The above-mentioned properties characterize the states in which porous materials can be encountered. In addition, this study also validates the sensitivity of wave propagation in porous materials and emphasizes the importance of carefully selecting the propagation medium and model to increase the relevance of this study to specific examples and the applications of these findings, such as environmental monitoring or geological exploration. In general, this study demonstrates that porous materials are highly sensitive to wave propagation. Furthermore, they can demonstrate the importance of the findings through experimental investigations in settings taken straight from the real world.

## Figures and Tables

**Figure 1 sensors-23-05108-f001:**
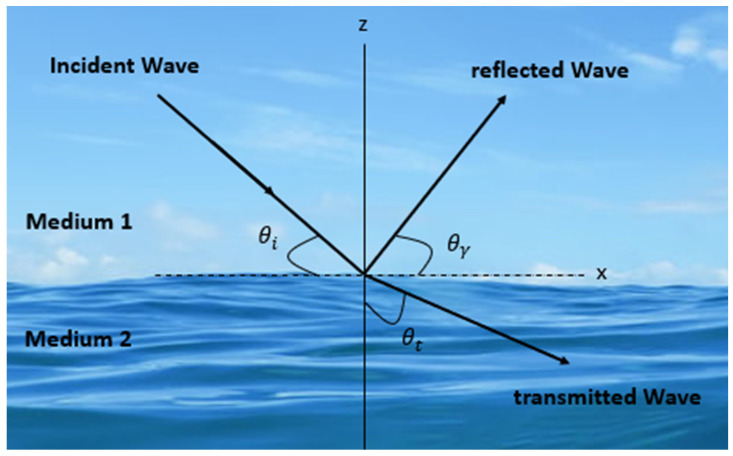
Wave propagation model.

**Figure 2 sensors-23-05108-f002:**
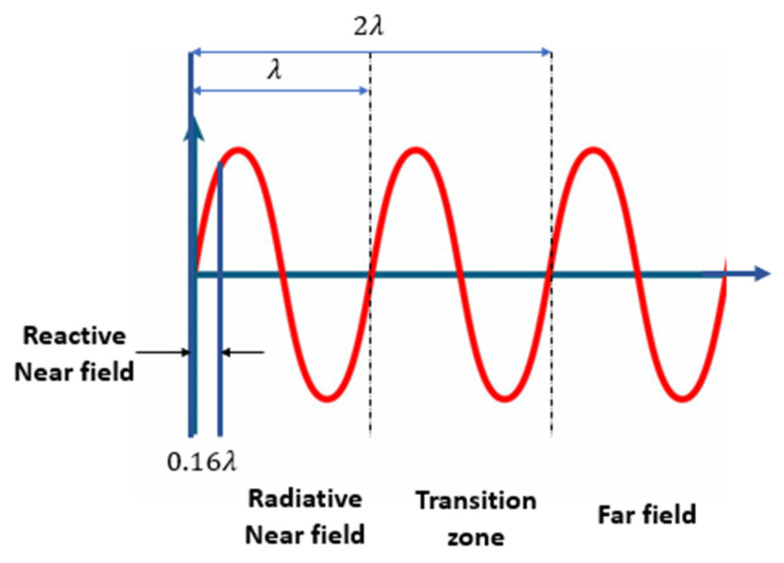
Interference of the acoustic waves.

**Figure 3 sensors-23-05108-f003:**
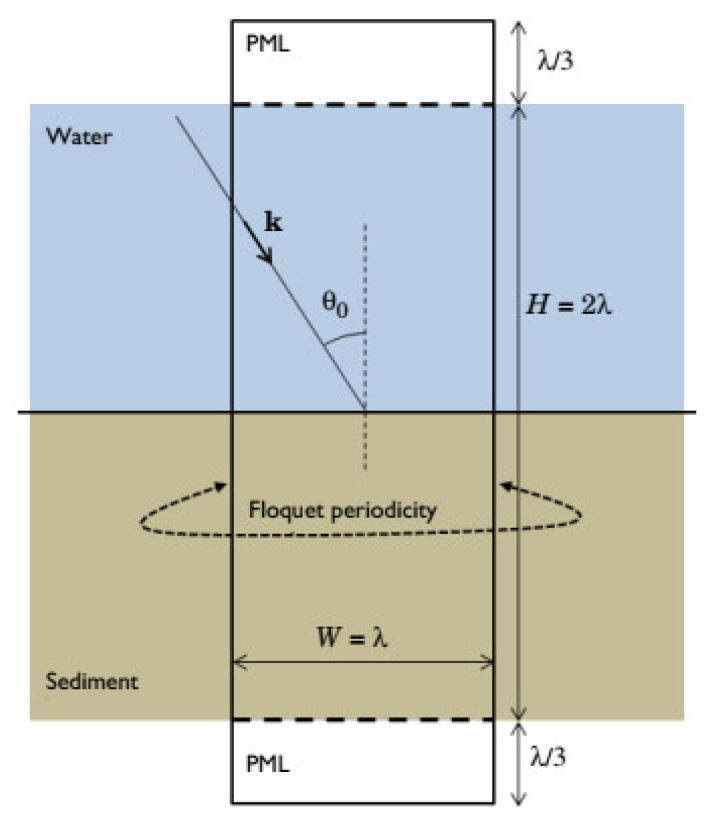
Current design model.

**Figure 4 sensors-23-05108-f004:**
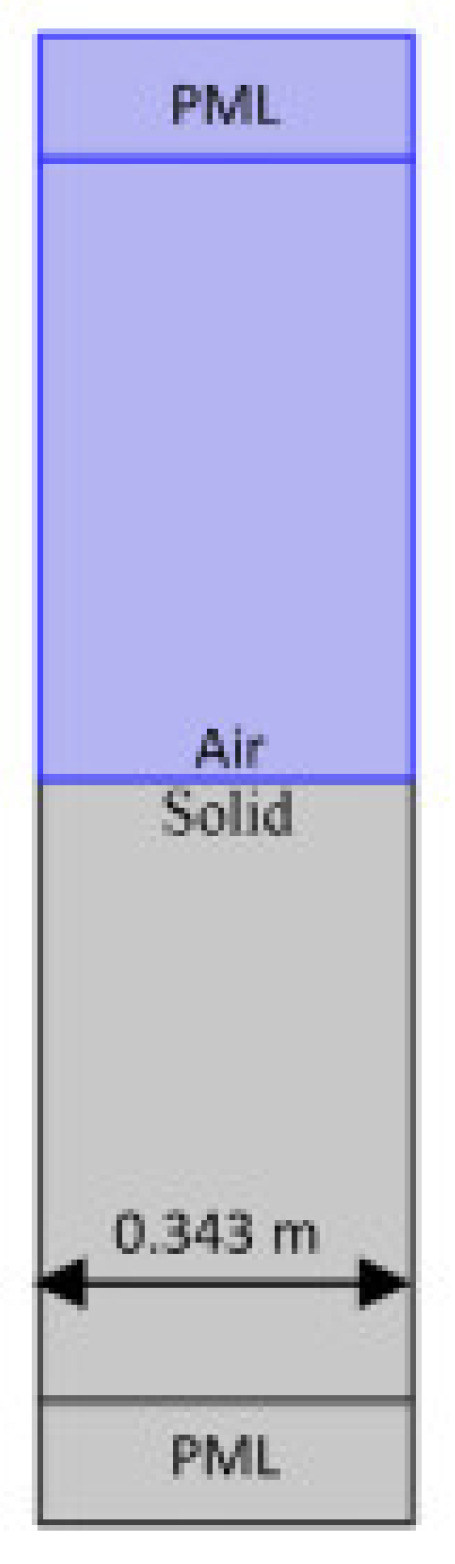
Geometrical modelling.

**Figure 5 sensors-23-05108-f005:**
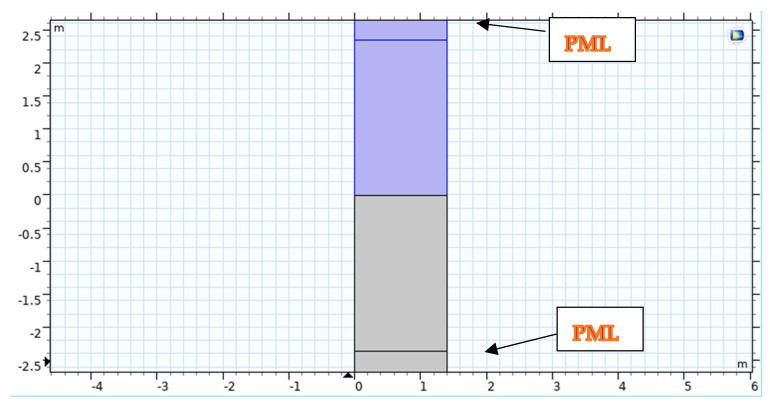
Simulation design model of UWSN.

**Figure 6 sensors-23-05108-f006:**
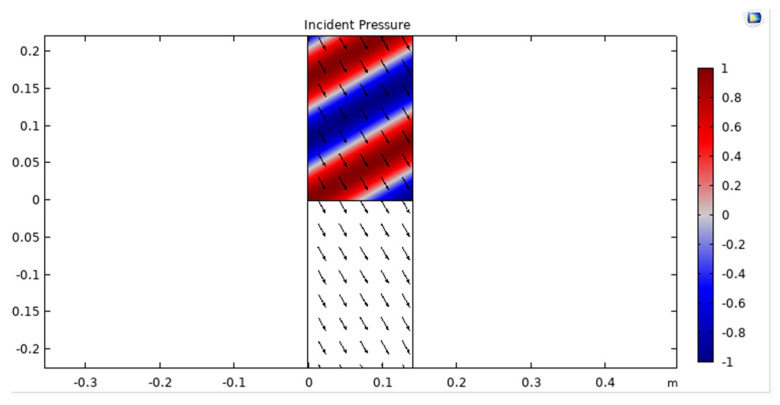
Incident pressure of acoustic wave in UWSN.

**Figure 7 sensors-23-05108-f007:**
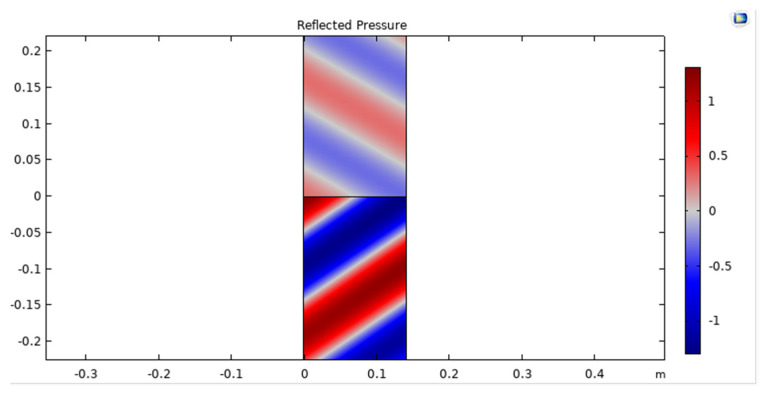
Reflected pressure of acoustic wave in UWSN.

**Figure 8 sensors-23-05108-f008:**
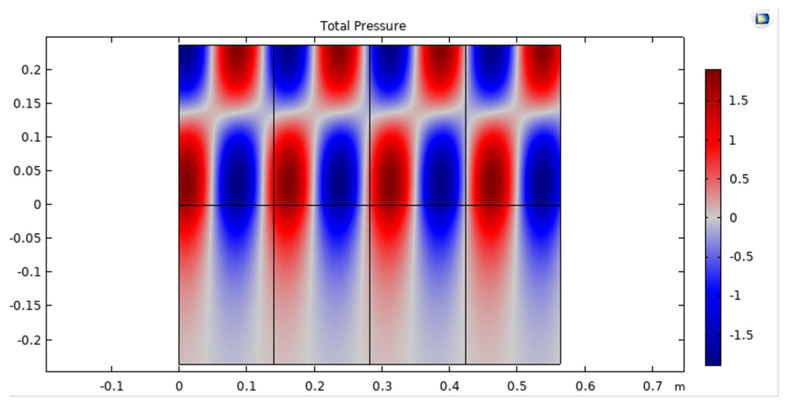
Total pressure of acoustic wave in UWSN.

**Figure 9 sensors-23-05108-f009:**
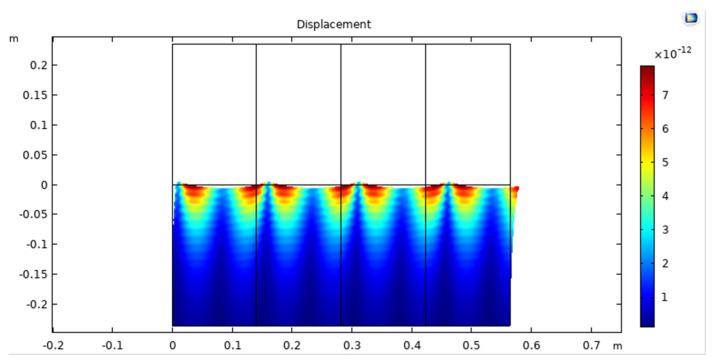
Displacement of acoustic wave in UWSN.

**Figure 10 sensors-23-05108-f010:**
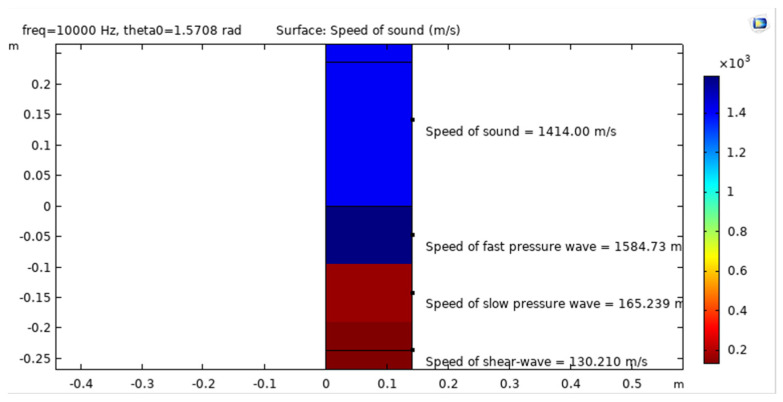
Acoustic wave speed in UWSN.

**Figure 11 sensors-23-05108-f011:**
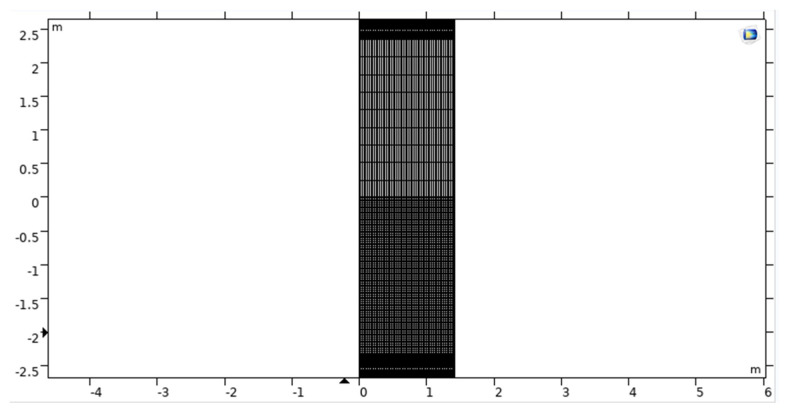
Mesh-view simulation model of UWSN.

**Figure 12 sensors-23-05108-f012:**
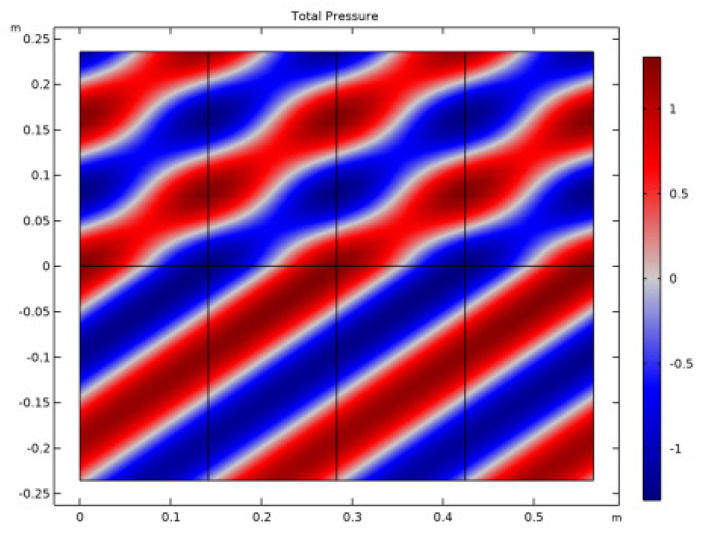
Periodic modelled system.

**Figure 13 sensors-23-05108-f013:**
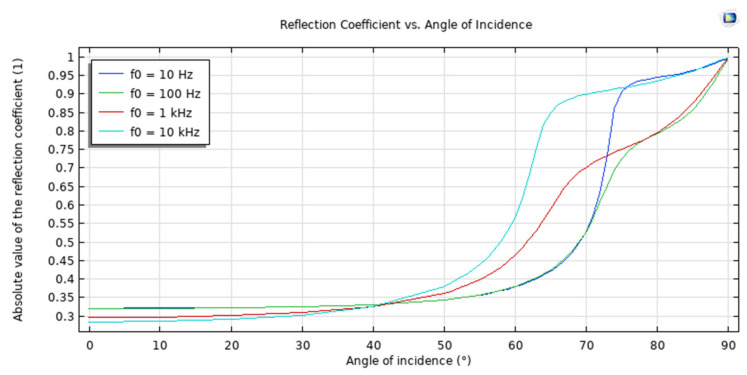
Reflection coefficient with inclination angle.

**Figure 14 sensors-23-05108-f014:**
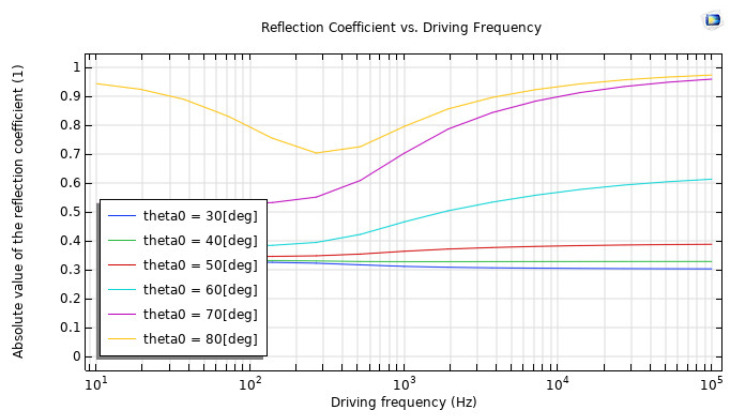
Driving frequency Reflection coefficient.

**Figure 15 sensors-23-05108-f015:**
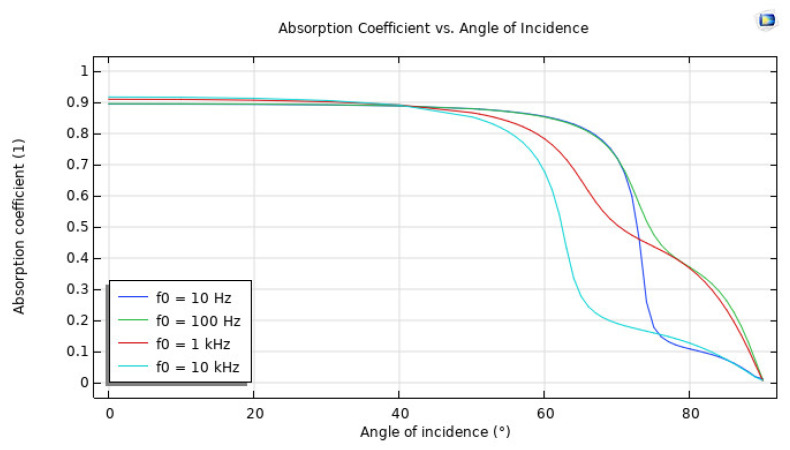
Absorption coefficient with inclination angle.

**Figure 16 sensors-23-05108-f016:**
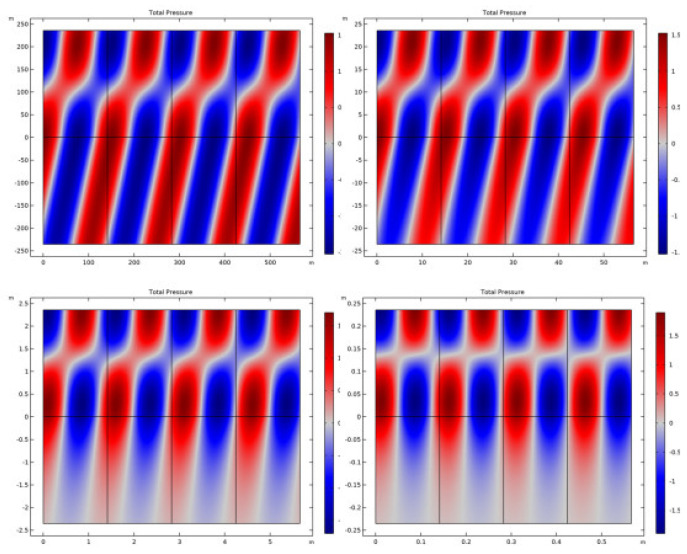
Frequency-dependent pressure.

**Figure 17 sensors-23-05108-f017:**
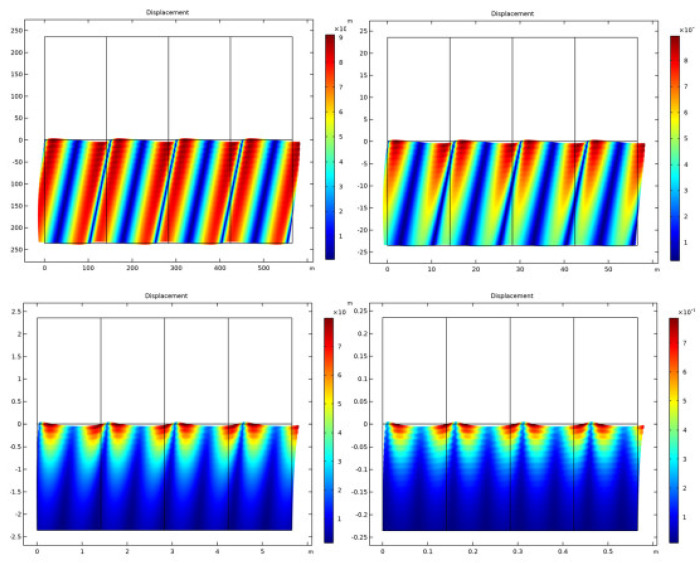
Frequency displacement.

**Table 1 sensors-23-05108-t001:** The essential parameters and their definitions.

Parameter	Values	Definition
f0	1000 [Hz]	Driving frequency
theta0	0 [deg]	Angle of incidence
c0	1414 [m/s]	Speed of sound in water
cs_poro	130 [m/s]	Speed of slow shear waves in the poroelastic wave domains
lam0	c0/f0	Wavelength at f0
rhoF	1000 [kg/m^3^]	Fluid density
Kf	rhoF × c0^2^	Bulk modulus of fluid
muF	1 × 10^−3^ [Pa·s]	Fluid viscosity
epsilonP	0.47	Porosity
a	4 × 10^−3^ [cm]	Pore size parameter
tau0	1.25	Tortuosity
Gc	G × (1 + i × logdec/pi)	Complex shear modulus of frame
W	lam0	Domain width
H	2 × lam0	Domain height
Hpml	lam0/3	PML height

## Data Availability

The datasets used during the current study are available from the corresponding author on reasonable request.

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
