# Peer review of "Acoustic Wave Reflection in Water Affects Underwater Wireless Sensor Networks"

_sensors, 2023, doi:10.3390/s23115108_

Round 1

Reviewer 1 Report

This work aims to measure the impact of physical qualities of materials on acoustic attenuation across a wide frequency range. Besides, they have generated reflection coefficient curves by adjusting the porousness and permeability of the poroelastic solid. Overall, this paper is interesting and timely. Here are some comments for this paper:

1. In the abstract section, the result of this work must be described briefly with data (please add some numerical results (achievements) in this part).

2. Underwater wireless sensor networks can be subject to interference from other underwater devices or environmental factors such as weather conditions or noise. How to fix those issues? 

3. Wireless sensor networks may be vulnerable to hacking or other security breaches, which could compromise sensitive data.

4. Many underwater wireless sensors are battery-powered, which means they have limited lifetimes and require frequent replacement or recharging.

5. AUV and sensors aided underwater information collection schemes are also the promising methods for maritime exploration. In particular, the optimization of data timeliness for IoUT is a challenging issue which becomes more severe in communication environment with relatively high transmission delay such as underwater communication environment. Please cite the latest related papers:(1) Average Peak Age of Information in Underwater Information Collection with Sleep-scheduling (2) Age-Optimal Information Gathering in Linear Underwater Networks: A Deep Reinforcement Learning Approach.

Author Response

Dear Editorial Board of MDPI Sensors

Thank you for reviewing our manuscript entitled Acoustic Wave Reflection in Water affects Underwater Wireless Sensor Networksby Kaveripakam Sathish, Monia Hamdi, Ravikumar Chinthaginjala Venkata, Mohammad Alibakhshikenari, Manel Ayadi, Giovanni Pau, Mohamed Abbas and Neeraj Kumar Shukla for publication in MDPI Sensors.  We are extremely thankful to the referees / reviewers and the editor of the journal for pointing out some important modifications needed in the manuscript. We have thoughtfully taken these comments into account. The changes/insertions done in the paper in responses to the reviewer’s/editor’s concerns are given point by point in the following paragraphs. 

We believe that the comments have been highly constructive and very useful to restructure our manuscript.  We also believe that the new data included in the research article really improved the quality of proposed feature sections of our manuscrip

Reviewer 2 Report

This paper studies the effect of the acoustic wave reflection in underwater wireless sensor networks.

The study and the method deserves some consideration. Furthermore, the paper writing is good in most of the parts. Generally, the technical contribution seems good. This paper meets the Sensors journal standards. 

Few comments are required to be addressed:

1- Although I believe that some core papers and methods are needed for the review part, I just wish if the authors could provide up-to-date references. In the current format, about 58% of the references were published more than 5 years ago. Indeed, five references (5, 6, 7, 14 and 19) were published more than 34 years ago. I don not see any need for such citation. Kindly, try to update some of them and I am sure you will find new interesting aspects regarding the topic.

2- The novelty of the paper has to be more clear. What is the new things  that this model brings in comparison to some of the above mentioned papers?

3- Are there a space for a compartive results with other existing models?

4- Could you provide some insightful and practical future research suggestions?

5- The entire references need a second revision in terms of their reference style.

Author Response

Dear Editorial Board of MDPI Sensors

Thank you for reviewing our manuscript entitled Acoustic Wave Reflection in Water affects Underwater Wireless Sensor Networksby Kaveripakam Sathish, Monia Hamdi, Ravikumar Chinthaginjala Venkata, Mohammad Alibakhshikenari, Manel Ayadi, Giovanni Pau, Mohamed Abbas and Neeraj Kumar Shukla for publication in MDPI Sensors.  We are extremely thankful to the referees / reviewers and the editor of the journal for pointing out some important modifications needed in the manuscript. We have thoughtfully taken these comments into account. The changes/insertions done in the paper in responses to the reviewer’s/editor’s concerns are given point by point in the following paragraphs. 

We believe that the comments have been highly constructive and very useful to restructure our manuscript.  We also believe that the new data included in the research article really improved the quality of proposed feature sections of our manuscript.

Reviewer 3 Report

In this manuscript, the authors try to measure the effect of material physical qualities on oblique incidence acoustic attenuation across a large frequency range. Generally speaking, the manuscript is easy to follow and the obtained results seem interesting. However, I have the following comments regarding the proposed work:

1-     The paper is more oriented toward application where we miss a clear research model in the work.

2-     The authors spent 6 pages in introduction section only to explain trivial concepts in UASN like wave propagation and interference. It is recommended to divide this section into several ones: Introduction, problem formulation, notations, etc.

3-     Please justify the selection of the parameter values shown in Table 1.

4-     Figures 6 to 9 are not properly described. The authors just write what observe in the figures without giving any explanation why we have these outcomes. Furthermore, Figure 8 shows an abnormal behavior when f0 = 10 Hz.

5-     A weakness point of this manuscript is that the authors did not compare their proposal design to the state of the art.  

Author Response

(The authors gave the same response as above.)

Round 2

Reviewer 1 Report

Most of my comments are addressed well, except that the author list of the reference [22] is wrong. Please resolve this problem before acceptance.

Author Response

(The authors gave the same response as above.)

Reviewer 3 Report

The authors addressed all my comments. I propose the paper for the publication.

Author Response

(The authors gave the same response as above.)
